# Cognitive Impairment Related to Chronic Kidney Disease Is Associated with a Decreased Abundance of Membrane-Bound Klotho in the Cerebral Cortex

**DOI:** 10.3390/ijms25084194

**Published:** 2024-04-10

**Authors:** María E. Rodríguez-Ortiz, Daniel Jurado-Montoya, Karen Valdés-Díaz, Raquel M. García-Sáez, Ana I. Torralbo, Teresa Obrero, Victoria Vidal-Jiménez, María J. Jiménez, Andrés Carmona, Fátima Guerrero, María V. Pendón-Ruiz de Mier, Cristian Rodelo-Haad, Antonio Canalejo, Mariano Rodríguez, Sagrario Soriano-Cabrera, Juan R. Muñoz-Castañeda

**Affiliations:** 1Nephrology Service, Reina Sofia University Hospital, Maimonides Institute for Research in Biomedicine of Cordoba (IMIBIC), University of Cordoba, Avda. Menéndez Pidal, s/n, 14004 Cordoba, Spain; maria.rodriguez@imibic.org (M.E.R.-O.); mvictoriaprm@gmail.com (M.V.P.-R.d.M.); crisroha@yahoo.com (C.R.-H.); marias.soriano.sspa@juntadeandalucia.es (S.S.-C.); juanr.munoz.exts@juntadeandalucia.es (J.R.M.-C.); 2Redes de Investigación Cooperativa Orientadas a Resultados en Salud (RICORS), Instituto de Salud Carlos III, 28029 Madrid, Spain; anatorralboromero@gmail.com (A.I.T.); carmonacarpio29@hotmail.com (A.C.); fatima.guerrero@imibic.org (F.G.); 3Maimonides Institute for Research in Biomedicine of Cordoba (IMIBIC), Reina Sofia University Hospital, University of Cordoba, Avda. Menéndez Pidal, s/n, 14004 Córdoba, Spain; daniel.jurado@imibic.org (D.J.-M.); valdeskc@gmail.com (K.V.-D.); rachel1506@gmail.com (R.M.G.-S.); teresa_18_4@hotmail.es (T.O.); vvidaljimenezv@gmail.com (V.V.-J.); maripijm83@gmail.com (M.J.J.); 4Department of Integrated Sciences/Research Center on Natural Resources, Health, and Environment (RENSMA), University of Huelva Campus el Carmen, Avda. Del Tres de Marzo, s/n, 21071 Huelva, Spain; antonio.canalejo@dbasp.uhu.es

**Keywords:** chronic kidney disease, cognitive impairment, bone and mineral disorder, Klotho

## Abstract

Cognitive impairment (CI) is a complication of chronic kidney disease (CKD) that is frequently observed among patients. The aim of this study was to evaluate the potential crosstalk between changes in cognitive function and the levels of Klotho in the brain cortex in an experimental model of CKD. To induce renal damage, Wistar rats received a diet containing 0.25% adenine for six weeks, while the control group was fed a standard diet. The animals underwent different tests for the assessment of cognitive function. At sacrifice, changes in the parameters of mineral metabolism and the expression of Klotho in the kidney and frontal cortex were evaluated. The animals with CKD exhibited impaired behavior in the cognitive tests in comparison with the rats with normal renal function. At sacrifice, CKD-associated mineral disorder was confirmed by the presence of the expected disturbances in the plasma phosphorus, PTH, and both intact and c-terminal FGF23, along with a reduced abundance of renal Klotho. Interestingly, a marked and significant decrease in Klotho was observed in the cerebral cortex of the animals with renal dysfunction. In sum, the loss in cerebral Klotho observed in experimental CKD may contribute to the cognitive dysfunction frequently observed among patients. Although further studies are required, Klotho might have a relevant role in the development of CKD-associated CI and represent a potential target in the management of this complication.

## 1. Introduction

Chronic kidney disease (CKD) constitutes a serious public health issue given its extraordinarily high prevalence. It is estimated that CKD will represent the fifth leading cause of death in the upcoming decades [1]. Furthermore, the onset and progression of CKD is accompanied by a variety of significant co-morbidities. Among these, cognitive impairment (CI), which may range from mild CI to dementia, stands out.

Overall, age is the main risk factor for the development of CI. Thus, after the age of 60, the risk for developing CI doubles each decade [2]. Chronic kidney disease is considered as a state of premature aging, including features such as vascular dysfunction and low-grade inflammation, among other disturbances [3]. The prevalence of CI progressively increases as renal function decreases [4], and it is estimated that the risk for cognitive dysfunction becomes 12% higher for each 10 mL/min/1.73 m^2^ decrease in the glomerular filtration rate [5]. In fact, CKD per se has been considered as a risk factor for developing CI and dementia [6].

Klotho, a protein expressed in the distal tubule of the kidney, was initially identified as an anti-aging molecule, and it was later related to the maintenance of mineral metabolism [7]. Thus, Klotho-deficient mice exhibit renal dysfunction, vascular calcification, fibrosis, osteopenia, pulmonary disease, and short lifespans. When it comes to renal function, a nephroprotective role has been attributed to Klotho, as it reduces renal fibrosis, inflammation, and oxidative stress [8]. In the context of CKD, the expression of renal Klotho progressively decreases as renal function declines. Hence, the reduction in Klotho is one of the hallmarks of the bone and a mineral disorder consequence of renal damage, which is associated with faster CKD progression and other negative extrarenal outcomes such as ectopic calcification, cardiomyopathy, or derangements in the levels of other molecules involved in the maintenance of mineral homeostasis [9].

The expression of Klotho has also been evidenced in the central nervous system [7], which would suggest that Klotho may influence brain functions. In fact, low levels of Klotho in serum have been associated with poor cognitive performance in the elderly [10], while high Klotho levels have been related to better scores when assessing cognitive function [11]. Altogether, this evidence suggests that Klotho contributes to better cognition. This observation has also been observed in a population with impaired renal function [12].

The connection between the abundance of Klotho in the central nervous system and cognitive performance has not been explored in detail. Furthermore, despite the high prevalence of CI in CKD and the fact that it entails a low quality of life, the interplay between CKD and CI is largely unknown.

In this study, our main goal was to assess the potential crosstalk between changes in neurobehavioral parameters and the levels of Klotho in the cerebral cortex of rats with CKD in comparison with animals with normal renal function.

## 2. Results

### 2.1. Plasma Biochemistry

The results obtained in plasma analytes related to mineral metabolism and renal function in both groups of animals are shown in Figure 1. The levels of plasma calcium and magnesium were not different between the healthy and uremic animals (10.42 ± 0.23 vs. 10.34 ± 0.73 mg/dL for calcium and 2.00 ± 0.09 vs. 2.31 ± 0.23 mg/dL for magnesium; Figure 1A,B). As expected, the rats with CKD exhibited hyperphosphatemia (4.08 ± 0.33 vs. 8.13 ± 1.68 mg/dL, *p* < 0.05; Figure 1C) and increased concentrations of plasma creatinine (0.77 ± 0.05 vs. 2.08 ± 0.28 mg/dL, *p* < 0.05; Figure 1D).

The circulating levels of the main hormones involved in the maintenance of mineral homeostasis were also assessed in this experimental model. Thus, the uremic animals had increased plasma PTH when compared with the healthy rats (229 ± 17 vs. 984 ± 348 pg/mL, *p* < 0.05; Figure 1E). In addition, renal dysfunction was associated with significant elevations in the plasma levels of both intact FGF23 (353 ± 58 vs. 7382 ± 2014 pg/mL, *p* < 0.05; Figure 1F) and c-terminal FGF23 (187 ± 34 vs. 3075 ± 858 pg/mL, *p* < 0.05; Figure 1G).

### 2.2. Urine Biochemistry

Table 1 shows the levels of the different analytes determined in the urine of the animals. The urinary excretion of calcium, adjusted by that of creatinine, was significantly higher in the rats with renal damage when they were compared with the controls (0.106 ± 0.030 vs. 0.025 ± 0.002, *p* < 0.05; Table 1). However, no statistical differences were found between the healthy and CKD rats when the excretion of phosphorus (0.22 ± 0.01 vs. 0.24 ± 0.03), magnesium (0.018 ± 0.004 vs. 0.019 ± 0.005), sodium (0.17 ± 0.04 vs. 0.24 ± 0.02), and potassium (0.53 ± 0.10 vs. 0.52 ± 0.06) were analyzed (Table 1).

### 2.3. Neurobehavioral Evaluation

Before the completion of the experiments, the animals in both experimental groups underwent two different cognitive tests.

A scheme of the object location task is depicted in Figure 2. Figure 3A shows the results of the animals in the phase test. In this graph, the white bars represent the exploration time of the animals with normal renal function, whereas the black bars show the data registered from the CKD group. Object A corresponds to the object that remained in the original position, whereas object B identifies the object whose location was changed. The time spent exploring object A was similar for both healthy and uremic rats (19 ± 3 vs. 22 ± 3 s; Figure 3A). The group of animals with normal renal function exhibited a normal exploratory behavior; hence, they spent significantly longer exploring and interacting with object B (47 ± 10 s, *p* < 0.05; Figure 3A). By contrast, the exploration time of objects A and B was similar in the group of animals with CKD (22 ± 3 vs. 16 ± 2 s; Figure 3A), with marked differences between the control and CKD rats in the time exploring object B (*p* < 0.01; Figure 3A).

We also found marked differences in the performance of both groups of rats in the light/dark test, which was intended to assess anxious/depressive behavior, as shown in Figure 3B. Although the normal rats tended to stay in the dark area for a slightly longer time, because it was considered a safe environment, this difference was not statistically significant with respect to the time spent in the light area (128 ± 8 vs. 172 ± 8 s; Figure 3B). However, the CKD animals avoided staying in the brightly illuminated area, thus suggesting a more marked anxious and depressive behavior (78 ± 8 vs. 222 ± 8 s, *p* > 0.05; Figure 3B).

### 2.4. Assessment of Klotho Abundance

The abundance of Klotho was evaluated at different locations. The circulating concentration of Klotho did not differ between the healthy and CKD rats (2.08 ± 0.07 vs. 2.01 ± 0.10 ng/mL; Figure 4A). However, the occurrence of renal dysfunction was associated with an elevation in the urinary excretion of Klotho (3.83 ± 0.24 vs. 2.40 ± 0.10 ng/mL in rats with normal renal function, *p* < 0.05; Figure 4B). This elevation remained significant in CKD when the concentration of urinary Klotho was corrected by urine creatinine (0.002 ± 0.001 vs. 0.010 ± 0.001 ng/mL, *p* < 0.05; Figure 4C).

The expression of Klotho was also evaluated in different tissues. In comparison with the normal rats, as it was expected, the abundance of Klotho in the renal tissue was reduced by 30% in the animals with renal failure (1.00 ± 0.03 vs. 0.70 ± 0.06-fold change vs. control, *p* < 0.05; Figure 4D).

Likewise, the expression of Klotho was evaluated in the frontal cortex. The expression of Klotho at this level was reduced by half in the animals with renal dysfunction when they were compared with the rats with normal renal function (1.00 ± 0.14 vs. 0.50 ± 0.04-fold change vs. control, *p* < 0.01; Figure 4E).

## 3. Discussion

The present study evaluated the cognition status and the expression of Klotho in the cerebral cortex of rats with CKD. We found that the uremic animals exhibited reduced levels of Klotho in the frontal cortex in comparison with the controls. The deficiency in the brain Klotho was accompanied by notable alterations in cognitive performance as assessed by neurobehavioral tests.

Chronic kidney disease was induced by the administration of a diet with a high content of adenine. The administration of dietary adenine is a widely used and validated experimental model of renal disease [13]. The amount of adenine in the diet may vary between 0.075% and 0.75%, producing minimal changes to severe renal damage. In our study, the rats were fed a diet containing 0.25% adenine. This diet has proven to induce structural and functional changes, which mimics many of the features observed in human renal disease, including those related to the disturbances in mineral homeostasis and the development of secondary hyperparathyroidism [14]. Thus, the group of animals that received dietary adenine had renal dysfunction, as evidenced by the plasma creatinine levels along with all of the hallmarks of the mineral disorder associated with CKD, i.e., markedly increased plasma levels of phosphorus, creatinine, PTH, and intact and c-terminal FGF23 and reduced renal Klotho expression.

The relationship between renal dysfunction and CI has been repeatedly reported. Clinical studies have suggested that patients with CKD have a higher risk of developing CI in comparison with the normal population [15]. In a longitudinal study, Murray et al. found an association between mild albuminuria and worse cognitive function and cognitive decline [16]. The CKD-REIN study showed a marked decrease in the score obtained in the Mini–Mental State Examination test per 10 mL/min/1.73 m^2^ reduction in the baseline-estimated glomerular filtration rate [17]. Similar results have been confirmed in a cohort of patients with CKD with type 2 diabetes [18]. In our study, cognitive function was evaluated by carrying out two different neurobehavioral tests. In both the object location task and the light/dark test, the CKD animals exhibited a differential performance when they were compared with the rats with normal renal function. Altogether, these results suggest that impaired cognition is associated with the presence of renal disease, and they are in agreement with previous studies in other preclinical models that were also performed in the context of CKD [19,20,21]. In fact, Wang et al. recently reported that uremic animals exhibit a progressive down-regulation of cognition-related markers as renal function declines, which is accompanied by impaired cognitive abilities [22].

In addition, we assessed, using immunohistological techniques, the expression of Klotho in the frontal cortex of the animals included in this study. Interestingly, the group of animals with renal dysfunction and altered neurobehavioral performance also showed decreased Klotho expression at this level. This finding may lead to the hypothesis that Klotho has a role in the development of CKD-associated cognitive dysfunction. In this regard, several lines of evidence point out that Klotho may be relevant for normal cognition. In the absence of CKD, the serum levels of Klotho have shown significant associations with the scores in tests intended to evaluate different cognition domains, such as the Mini–Mental State Examination, the Clinical Dementia Rating, or the Digit Symbol Substitution test [23,24], while low concentrations of Klotho have been related to worse cognition [10]. In line with these observations, Klotho overexpression has been evidenced to improve cognitive function in nonhuman primates [25]. There is a scarcity of studies on this relationship in the context of CKD, but a recent study performed with patients with CKD in stages 1–5 support the notion that derangements in Klotho might be involved in the development of CKD-associated cognitive dysfunction [12]. In the preclinical context, Degaspari et al. found altered soluble Klotho in the brain and NF-κB-TNFα signaling in rats after 120 days of renal damage induced by 5/6 nephrectomy [26]. Here, we report that changes in Klotho expression can be found earlier than the progression of CKD. In addition, our results show that these changes are produced on cerebral transmembrane Klotho. Also, we show that reduced Klotho is associated with high serum phosphorus and elevated FGF2. It may be that inflammation mediators play an important role in CI, but the key issue is whether a reduction in the phosphorus load may help maintain brain Klotho expression, improving CI.

Several elements proposed as potential contributors are related to Klotho abundance, which further supports the idea of the relevance of Klotho in CI. In particular, high protein intake has been associated with a higher risk of CI [27]. The consumption of large amounts of protein may contribute to hyperphosphatemia in CKD, which is known to down-regulate Klotho levels [28]. Several studies have suggested a potential relationship between dyslipidemia and cognitive dysfunction [29,30], although not in CKD, and notably, the NHANES study reported an inverse relationship between Klotho and dyslipidemia [31]. In addition, uremic toxins have been suggested to contribute to CI by inducing metabolic changes that eventually lead to apoptotic death in neuronal cells [32]; in parallel, uremia has been demonstrated to contribute to Klotho down-regulation through epigenetic mechanisms [33]. Finally, an association has been described between the score obtained in the Montreal Cognitive Assessment Test and inflammatory parameters in predialysis patients. In this regard, it should be considered that CKD is regarded as an inflammatory state, and the direct contribution of inflammation to reduced Klotho expression has been shown in different experimental works [34,35]. It should be noted that although the interplay between Klotho and inflammation has been reported to occur at the renal level, it cannot be discarded that similar effects may also occur in the brain, which is a topic worth investigating in future studies.

In sum, all of this evidence supports the notion that the down-regulation of Klotho may play a role in the development and progression of CI associated with renal disease. Nevertheless, it seems quite unlikely that the deficiency in Klotho abundance at the brain level is uniquely responsible in the pathogenesis of CKD-associated cognitive deficiency. Thus, it cannot be ruled out that other factors associated with alterations in the cerebral blood flow, often related to dialysis [36,37,38], or the impaired integrity of the blood–brain barrier [39] may also contribute to the development of CI in the setting of CKD. Interestingly, lifestyle-related factors [40] and the educational levels of patients [41,42] have also been linked to cognitive dysfunction in this context. Nevertheless, the contribution of all of these elements requires further investigation.

The relevance of CKD-related cognitive dysfunction partly relies on the large proportion of patients affected. According to different studies, the prevalence of mild CI in CKD ranges from 27 to 62%, and it eventually may progress to dementia. The prevalence of dementia among patients undergoing hemodialysis, peritoneal dialysis, and kidney transplant is in the ranges of 8–37%, 4–33%, and 7–22%, respectively, while the percentage of patients affected by dementia in the general population is around 5% [6]. Cognitive dysfunction affects domains such as memory, executive function, attention, orientation, visuospatial ability, and language ability. Thus, it is not surprising that at least part of the relevance of CKD-associated cognitive dysfunction relies on lower treatment compliance [43] along with the worse quality of life experienced by these patients. In this sense, very recent studies have reported apathy symptoms in patients with CKD and lower scores in terms of physical function, social and emotional function, and mental health, among other parameters [24,44].

Despite the findings described here on CKD-associated cognitive impairment, several limitations should be acknowledged. First, the experiments only involved male rats, which may have introduced gender bias given that the male gender is associated with a faster progression of renal disease [45] and, possibly, CKD-associated complications. Second, these are preclinical results that should be validated in a clinical setting; however, the availability of samples from brain tissue constitutes an issue to overcome. On the other hand, it would be of great interest to test whether Klotho supplementation might prevent or reverse impaired cognitive function in experimental CKD.

In conclusion, the results shown here explore the features of cognitive dysfunction related to CKD. We demonstrated that, in the setting of CKD, along with the classic features of renal disease, there is a decrease in cognitive abilities, which appear to be associated with a marked reduction in the expression of Klotho in the cerebral cortex. Undoubtedly, many aspects of CKD-related CI require further investigation. Experimental work is needed to identify the factors and mechanisms underlying the development and progression of cognitive dysfunction. In addition, clinical studies are required to determine whether the cognitive dysfunction observed in CKD exhibit specific patterns and to implement valid methods for cognitive assessment in routine clinical practice as well as biomarkers that are capable of identifying patients who have a high risk of developing CI. Gaining insight into this issue will allow us to define therapeutic targets to prevent or slow down CKD-associated cognitive deficiency.

## 4. Material and Methods

### 4.1. Animals

Experimental work with animals was carried out at the animal facility of the University of Cordoba. Male Wistar rats (Charles River Laboratories, Wilmington, MA, USA) that were 9–10 weeks old and weighing 280–300 g were individually housed using a 12 h/12 h light/dark cycle and given ad libitum access to a breeding diet (Altromin GmbH, Lage, Germany). Ethics approval was obtained from the Ethics Committee for Animal Research of the University of Cordoba. Animals received humane care in compliance with the Principles of Laboratory Animal Care formulated by the National Society for Medical Research and the 2010/63/EU Directive.

### 4.2. CKD Experimental Model

Chronic kidney disease was induced by the administration of an adenine-enriched diet containing 0.25% adenine, normal amounts of calcium (0.6%), and an elevated content of phosphorus (0.9%). A group of healthy rats that was used as a control group received a standard diet with normal amounts of calcium (0.6%) and phosphorus (0.6%). Both diets were purchased from Altromin. Animals (five rats per experimental group) were kept on these diets for six weeks. Before sacrifice, rats were housed in metabolic cages to collect 24 h urine samples.

### 4.3. Neurobehavioral Tests

In the two days before sacrifice, animals underwent different tests. The equipment for the neurobehavioral tests was obtained from Harvard Apparatus/Panlab (Holliston, MA, USA). In the *object location task* (scheme depicted in Figure 2), intended for the assessment of memory and exploratory behavior, a habituation phase was first carried out. In this phase, animals were allowed to freely explore the arena for ten minutes for seven days. The familiarization phase was performed the following day; in this phase, two objects were placed in particular positions in the arena, and animals could explore them for ten minutes. These objects had identical sizes, colors, and shapes. The testing phase, during which one of the objects was moved to a different position, was performed three hours later, and again, animals were allowed to explore for ten minutes. Prior to performing this test, a validation of the objects was carried out in a different group of animals that were not included in this study to ensure that there was no preference for a particular pair of objects or a position in the arena.

The *light/dark test* was performed for the evaluation of anxious/depressive behavior. This task is based on the innate aversion of rodents to excessively illuminated areas along with their tendency to show spontaneous behavior to novel environments. It was carried out in a device divided into two compartments, one of them being brightly illuminated (aversive area) and another one darkened (safe area), with an opening between both of them. Thus, the avoidance of the lit area is indicative of anxious behavior. To address this issue, the time that the animals spent in each compartment within five minutes was recorded.

### 4.4. Serum and Urine Biochemistries

At the end of the experiment, rats were sacrificed by aortic puncture and exsanguination under general anesthesia. Blood was collected in heparinized syringes, and plasma was separated by centrifugation and stored at −80 °C until assayed. To determine the levels of the molecules of interest at the moment of sacrifice, ELISA kits were used for the determination of plasma intact PTH (Quidel, San Diego, CA, USA), intact FGF23 (Kainos Laboratories, Tokyo, Japan), c-terminal FGF23 (Quidel), and urine and plasma Klotho (BT Lab, Shanghai, China). Serum and urine calcium, phosphorus, magnesium, and creatinine levels were quantified by spectrophotometry (Biosystems, Barcelona, Spain). The urinary excretion of sodium and potassium was determined with the aid of a Spotlyte Na/K analyzer (Menarini Diagnostics, Barcelona, Spain).

### 4.5. Immunohistochemistry

The tissues collected after sacrifice were processed as follows: Kidneys were fixed in 4% formalin and embedded in paraffin. The brains were embedded in OCT embedding compound (Sakura FineTek, Torrance, CA, USA). Three-micrometer sections of both tissues were hydrated prior to performing immunohistochemical staining. Renal sections were also deparaffined before hydration. For this purpose, the ImmPRESS^®^ HRP Universal PLIS Polymer Kit (Vector Laboratories, Newark, CA, USA) was used. First, renal sections were microwave-treated in 0.01 mmol/L citrate buffer (pH 6.0) for 20 min. Then, both tissues followed the same protocol. Sections were incubated in BLOXALL^®^ Endogenous Enzyme Blocking Solution for 10 min to inactivate endogenous peroxidase. Then, they were blocked with 2.5% horse serum for 20 min at room temperature in a humidified chamber to avoid non-specific binding. After this step, tissue was incubated overnight at 4 °C with anti-Klotho antibody (1:200 dilution, Alpha Diagnostic, San Antonio, TX, USA) in a humidified chamber. After rising, the slides were incubated with peroxidase-labeled polymer conjugated with anti-mouse/rabbit IgG for 30 min. Then, sections were treated with 3,3′-diaminobenzidine-tetrachloride chromogen solution for 1 min. Every step was followed by two washes with phosphate-buffered saline (Sigma-Aldrich, Saint Louis, MO, USA). Sections were counterstained with hematoxylin (PanReac AppliChem ITW reagents, Darmstadt, Germany) for 1 min and dehydrated. Staining with polymer without primary antibody was carried out as negative control to avoid unspecific signals.

### 4.6. Statistical Analysis

Statistical analyses between both experimental groups were assessed using the *t* test or the corresponding non-parametric test. A *p*-value inferior to 0.05 was considered statistically significant. Statistical analyses were performed using SPSS 15.0 statistical software (SPSS Inc., Chicago, IL, USA).

## Figures and Tables

**Figure 1 ijms-25-04194-f001:**
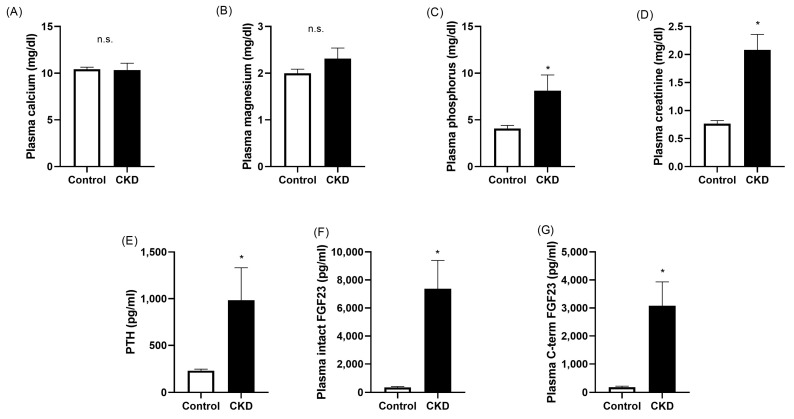
The plasma biochemistry of the animals included in this study. The white and black bars represent the data obtained from the control and CKD groups, respectively. The animals with renal dysfunction had similar levels of calcium (**A**) and magnesium (**B**). The circulating concentrations of phosphorus (**C**), creatinine (**D**), PTH (**E**), intact FGF23 (**F**), and c-terminal FGF23 (**G**) were significantly increased in the groups of animals with renal dysfunction. CKD, chronic kidney disease. n.s., non-significant; * *p* < 0.05 vs. control.

**Figure 2 ijms-25-04194-f002:**
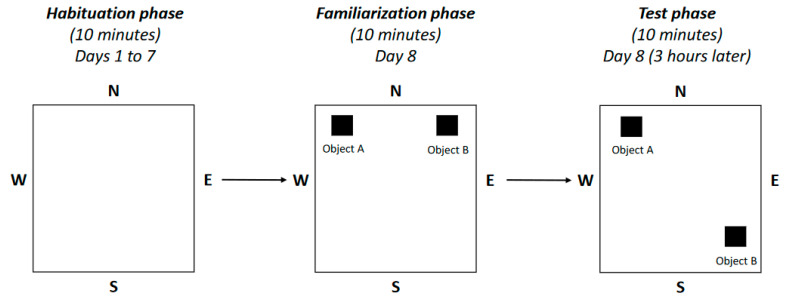
A scheme of the object location task. This test consisted of three different phases. In the habituation phase, the animals explored the arena for ten minutes during seven days. In the familiarization phase, two identical objects were placed in certain positions in the arena, and the animals could explore them for ten minutes. In the testing phase, which was performed three hours after the familiarization phase, object B was changed to a different position and, again, the animals were allowed to explore them for ten minutes. The time spent exploring both objects was recorded for each animal. N, north; S, south; E, east; W, west.

**Figure 3 ijms-25-04194-f003:**
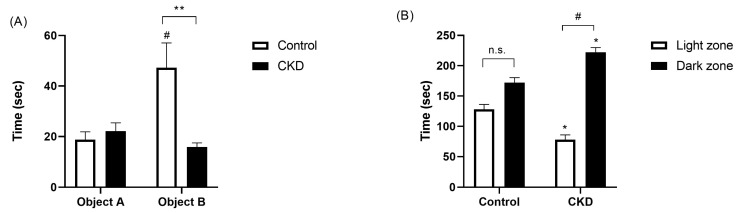
An assessment of cognitive function. (**A**) represents the results obtained by the experimental groups in the object location task. There were no differences in the time spent by both groups exploring object A. However, the CKD animals (black bars) spent significantly less time exploring object B in comparison with the healthy rats, which was indicative of an impaired behavior. ** *p* < 0.01 vs. control; ^#^
*p* < 0.05 vs. object A. (**B**) shows the performance of the animals in the light/dark test. The rats with normal renal function spent statistically equal time in the light (white bars) and the dark (black bars) zones, whereas the animals with renal dysfunction stayed in the dark zone longer, as it was considered a safe environment. CKD, chronic kidney disease. * *p* < 0.05 vs. control; n.s., non-significant; ^#^
*p* < 0.05 vs. light zone.

**Figure 4 ijms-25-04194-f004:**
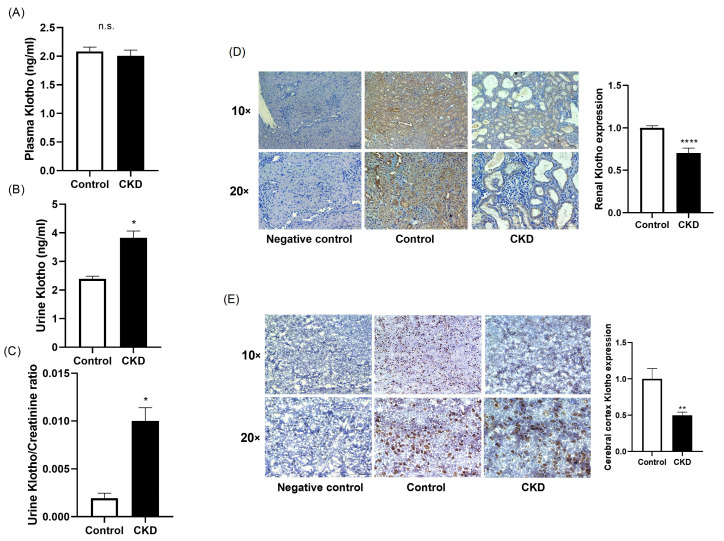
The determination of the levels of Klotho in the experimental groups. The circulating concentration of Klotho did not differ between the controls and CKD animals (**A**). However, the concentration of Klotho in urine was substantially higher in the rats with renal damage (**B**), and such difference persisted when the concentration of Klotho was adjusted by that of creatinine (**C**). As expected, the rats with CKD had a marked reduction in renal Klotho. Magnification, 10× and 20× (**D**). When the expression of Klotho was evaluated in the cerebral cortex, a reduction of 50% was observed in the animals with renal dysfunction. Magnification, 10× and 20× (**E**). CKD, chronic kidney disease. n.s., non-significant; * *p* < 0.05 vs. control; ** *p* < 0.01 vs. control; **** *p* < 0.0001 vs. control.

**Table 1 ijms-25-04194-t001:** Urine analytes in control and CKD groups.

	Control (*n* = 5)	CKD (*n* = 5)
Calcium/creatinine ratio	0.025 ± 0.002	0.106 ± 0.030 *
Phosphorus/creatinine ratio	0.22 ± 0.01	0.24 ± 0.03
Magnesium/creatinine ratio	0.018 ± 0.004	0.019 ± 0.005
Sodium/creatinine ratio	0.17 ± 0.04	0.24 ± 0.02
Potassium/creatinine ratio	0.53 ± 0.10	0.52 ± 0.06

CKD, chronic kidney disease. * *p* < 0.05 vs. control.

## Data Availability

The data that support the findings of this study are available from the corresponding author (M.R.) upon reasonable request.

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
