# Peer review of "Cognitive Impairment Related to Chronic Kidney Disease Is Associated with a Decreased Abundance of Membrane-Bound Klotho in the Cerebral Cortex"

_ijms, 2024, doi:10.3390/ijms25084194_

Round 1
Reviewer 1 Report
Comments and Suggestions for Authors
Maria et al. Using rats, they found that Klotho is associated with cognitive impairment (CI) in chronic kidney disease (CKD). The methods employed in their study are commendable for their logical approach and are closely consistent with data previously collected in my own laboratory. On this basis, I tend to believe that the experimental data provided are credible. Nonetheless, I find the scope of this study somewhat limited and question its suitability for publication in the IJMS due to lack of novelty.
As if the paper's main novelty wasn't enough, it focused on the well-documented downregulation of Klotho - a key protein known for protecting the kidneys - in CKD, and this related to CI. The literature already contains substantial evidence that Klotho is significantly down-regulated in the context of CKD. Furthermore, the role of Klotho in influencing CI is a well-established phenomenon. The link between Klotho and CI in CKD has attracted considerable attention, particularly Degaspari's 2015 article in PLoS One, which not only explored this relationship but also related it to NF-kB and Potential mechanisms linking the TNF-a pathway. Subsequent studies, such as Free Radical Biology and Medicine by Nguyen et al., 2022, introduced new findings by elucidating the interplay between the Nrf2/GPx-1/ERK/CREB pathway and its mechanisms. In contrast, the current paper fails to advance our understanding of the mechanisms involved and essentially restates established knowledge.
Author Response
Manuscript ID: ijms-2919099
“Cognitive impairment related to chronic kidney disease is associated with decreased abundance of membrane-bound Klotho in cerebral cortex”
Reviewer #1
Maria et al. Using rats, they found that Klotho is associated with cognitive impairment (CI) in chronic kidney disease (CKD). The methods employed in their study are commendable for their logical approach and are closely consistent with data previously collected in my own laboratory. On this basis, I tend to believe that the experimental data provided are credible. Nonetheless, I find the scope of this study somewhat limited and question its suitability for publication in the IJMS due to lack of novelty.
As if the paper's main novelty wasn't enough, it focused on the well-documented downregulation of Klotho - a key protein known for protecting the kidneys - in CKD, and this related to CI. The literature already contains substantial evidence that Klotho is significantly down-regulated in the context of CKD. Furthermore, the role of Klotho in influencing CI is a well-established phenomenon. The link between Klotho and CI in CKD has attracted considerable attention, particularly Degaspari's 2015article in PLoS One, which not only explored this relationship but also related it to NF-kB and Potential mechanisms linking the TNF-a pathway. Subsequent studies, such as Free Radical Biology and Medicine by Nguyen et al., 2022, introduced new findings by elucidating the interplay between the Nrf2/GPx-1/ERK/CREB pathway and its mechanisms. In contrast, the current paper fails to advance our understanding of the mechanisms involved and essentially restates established knowledge.
We sincerely appreciate the reviewer's commentary and recognize a significant oversight in the preparation of our manuscript regarding the omission of this pivotal work. Regrettably, we acknowledge our failure to engage with this literature. Consequently, we fully comprehend the reviewer's observations and concur that our work may indeed lack novelty. However, we proceed to delineate the reasons why we maintain that the existence of Degaspari et al.'s work does not invalidate the publication of our findings in IJMS.
Firstly, one of the primary distinctions lies in the different experimental models employed. While Degaspari utilizes a chronic model involving nephrectomized animals observed over a span of 120 days, our study employs an adenine-induced model, assessing changes within a considerably shorter timeframe (42 days). By presenting these results within a truncated period, we aim to underscore the notion that the decline in cerebral Klotho levels may serve as a causal factor in cognitive deterioration rather than a consequence.
Secondly, Degaspari et al. conducted measurements of soluble Klotho derived from hippocampal and frontal cortex homogenates. As it is known, soluble Klotho can be produced via two pathways: either through the cleavage of transmembrane Klotho or via synthesis from an isoform excluding the transmembrane and cytosolic domains. Thus, this soluble Klotho is secreted into the bloodstream and may bind to specific receptors in target organs. Degaspari's work measures soluble Klotho and does not preclude the possibility that the presence of this Klotho in the brain may stem from an extracerebral source of Klotho. In contrast, our study introduces the novelty of demonstrating immunohistochemistry data using an antibody recognizing membrane-bound Klotho. Consequently, our work presents the novelty that in a renal injury model different from nephrectomy and within a much earlier timeframe than that shown by Degaspari, a decrease in membrane-bound Klotho produced in the brain and kidney is documented.
Also, we show that reduced Klotho is associated with high serum P and elevated FGF23, and it is interesting to observe that our renal failure animal presents Klotho deficiency. It may be that inflammation mediators play an important role in CI, but a key issue is whether a reduction in phosphate load is an important factor to maintain tissue Klotho and improve cognitive dysfunction. In previous works we have shown that reducing intake of phoshorus helps control FGF23 levels and maintain renal Klotho (Rodríguez-Ortiz et al. Clin Sci 2020; 134:15-32). Thus, a question that remains to be answered is whether we could preserve brain Klotho by reducing P load in CKD.
Accordingly, we have tried to incorporate this information in the discussion of the manuscript (p. 7): “In addition, we have assessed by immunohistological techniques the expression of Klotho in the cerebral the frontal cortex of the animals included in the study. […] As far as we know, tThis finding has not been previously described and may lead to hypothesize a role of Klotho in the development of CKD-associated cognitive dysfunction. […] In the preclinical context, Degaspari et al. found altered soluble Klotho in brain and NF-kB-TNFa signaling in rats after 120 days of renal damage induced by 5/6 nephrectomy [26]. Here we report that changes in Klotho expression can be found earlier over the progression of CKD. In addition, our results show that these changes are produced on cerebral transmembrane Klotho Also, we show that reduced Klotho is associated with high serum phosphorus and elevated FGF2. It may be that inflammation mediators play an important role in CI, but the key issue is whether a reduction of phosphorus load may help maintain brain klotho expression improving CI”.
On the other hand, the title of the manuscript has been slightly changed to “Cognitive impairment related to chronic kidney disease is associated with decreased abundance of membrane-bound Klotho”, which may be more appropriate according to the results shown in the article.

Reviewer 2 Report
Comments and Suggestions for Authors
Dear authors, thank you very much for giving me the opportunity to read your work. You describe the effects of kidney failure on cognitive function of Wistar rats. You state that there is an association of diminished Klotho expression, renal function deterioration and cognitive function.
I have the following comments/suggestions :
Chronic kidney disease evolves through time. The average lifespan of the Wistar rat is about 2-3 years. This simulates about 80 years of the human adult life. Adenine diet effects are reversible once one withholds diet.
It is not clear to me how long did you observe the aformentioned changes (weeks, months?) while on adenine diet. Please specify within the materials and methods section. Another point is whether you had the opportunity to test rats while withholding adenine diet. Which was the effect on Klotho and cognitive function tests?
Which was the rationale on utilizing male only Wistar rats? Please specify?
Which area of the cerebral cortex was examined? Was there any differential expression observed in the cerebral cortex? Did you have alternative methods (e.g. WB, or RT - PCR) on quantifying Klotho expression in the cerebral cortex?
Is this a model of chronic kidney disease or a model of acute kidney disease? This question is relevant since adenine causes kidney damage in a temporal manner.
Minor: Please improve image quality.
All the best.
Author Response
Manuscript ID: ijms-2919099
“Cognitive impairment related to chronic kidney disease is associated with decreased abundance of membrane-bound Klotho in cerebral cortex”
Reviewer #2
Dear authors, thank you very much for giving me the opportunity to read your work. You describe the effects of kidney failure on cognitive function of Wistar rats. You state that there is an association of diminished Klotho expression, renal function deterioration and cognitive function.
I have the following comments/suggestions:
Chronic kidney disease evolves through time. The average lifespan of the Wistar rat is about 2-3 years. This simulates about 80 years of the human adult life. Adenine diet effects are reversible once one withholds diet.
It is not clear to me how long did you observe the aformentioned changes (weeks, months?) while on adenine diet. Please specify within the materials and methods section. Another point is whether you had the opportunity to test rats while withholding adenine diet. Which was the effect on Klotho and cognitive function tests?
We thank the reviewer for these comments. We regret that the timing of the changes described has not been sufficiently detailed on the manuscript. Animals were on the adenine-enriched diet for six weeks (abstract and p. 9). The neurobehavioral tests were carried out in the days immediately prior to sacrifice and the biochemical and immunohistochemical determinations were performed in the plasma samples and tissues collected when animals were sacrificed, and the urine samples collected after housing of the animals in the metabolic cages the day before the sacrifice.
To clarify these points, the following information has been added in the corresponding sections of Material and methods:
Section 4.3. Neurobehavioral tests, p. 9: “Before sacrifice In the two days before sacrifice, animals underwent two different tests […]”
Section 4.4. Serum and urine biochemistries, p. 9: “[…]To determine the levels of the molecules of interest at the moment of sacrifice, ELISA kits were used for the determination of plasma intact PTH (Quidel, San Diego, CA, U.S.A.), intact FGF23 (Kainos Laboratories, Tokyo, Japan), c-terminal FGF23 (Quidel), and urine and plasma Klotho (BT Lab, Shangai, China) […]”.
Section 4.5. Immunohistochemistry, p. 9: “The tissues collected after sacrifice were processed as it follows. Kidneys were fixed in 4% formalin and embedded in paraffin […]”.
Regarding the second question, in this particular study we did not examine the effect of withholding the high adenine diet or, in another words, to determine whether cognitive impairment may be reversed by removing the renal-damaging stimulus. The renal damage induced as a consequence of adenine administration occurs in a dose- and time-dependent manner. According to the findings published by Okada et al., “Reversibility of adenine-induced renal failure” (Clin Exp Nephrol 1999; 3: 82-88), CKD becomes irreversible in this experimental model when plasma creatinine levels exceed 1.8 mg/dl. However, it should be taken into account that in the study by Okada, this concentration of creatinine was achieved by administering a diet containing 0.75% adenine for two weeks. Although the settings established in our study are different and the amount of adenine contained in the diet was 0.25%, the concentration of plasma creatinine in our CKD group was superior to 2.0 mg/dl, probably because our diet also contained high phosphorus (0.9%) to further promote renal damage. If we consider that the threshold of creatinine is crucial for the reversal of renal damage, we may presume that CKD animals might have not recovered even after adenine withholding. If this was the case, theoretically both Klotho levels and cognitive performance would be similar after adenine removal. In this regard, a very recent work has summarized the current evidence on Klotho-increasing strategies, including hormonal agents, RAS inhibitors, anti-inflammory agents, statins, vitamin D receptor agonist among others (Poursistany et al. “The current and emerging Klotho-enhacement strategies”. Biochemical and Biophysical Research Communications 2024; 22:693:149357). However, it remains to be investigated whether these therapies might be effective promoting cognitive performance. Nevertheless, our group has shown (Santamaria et al. Sci Rep 2018; 8:13701) in renal failure rats that for a given reduction in the GFR the available Klotho can be increased by reducing the amount of phosphorus in the diet, which suggest that there are strategies to optimize the levels of tissue and blood Klotho in renal failure. Undoubtedly, future studies addressing a possible reversion and/or prevention of cognitive impairment related to renal disease are warranted.
Which was the rationale on utilizing male only Wistar rats? Please specify?
We thank the reviewer for this question that may be indeed an interesting point. It should be remarked that male and female sex hormones differentially influence kidney damage through inflammation, oxidative stress, RAS system, endothelin system, energy metabolism, or fibrosis. Overall, male gender is related with worse CKD progression, while a renoprotective effect has been attributed to female hormones. Thus, we chose to perform the experiments under the a priori most favorable conditions for the progression of renal disease and cognitive impairment. Therefore, subsequent studies should be conducted to assess these differences in both genders, investigating potential differences and correlations with other parameters of renal damage. In this sense, we acknowledge that only using Wistar rats might introduce gender bias in these experiments Accordingly, this limitation has been acknowledged in the manuscript (p. 8): “Despite the novel findings described here on the CKD-associated cognitive impairment, several limitations should be acknowledged. First, the experiments only involved male rats, which may have introduced gender bias given that male gender is associated with faster progression of renal disease [45] and, possibly, also that of the CKD-associated complications. Second, First, these are preclinical results that should be validated in the clinical setting; […]”
Which area of the cerebral cortex was examined? Was there any differential expression observed in the cerebral cortex? Did you have alternative methods (e.g. WB, or RT - PCR) on quantifying Klotho expression in the cerebral cortex?
We greatly appreciate the reviewer's comment. However, we extracted the entire brain from the animals and promptly embedded it in OCT resin and froze it. Unfortunately, we do not have additional samples to measure Klotho by using any other technique. On the other hand, positive staining for Klotho was evident in the region of the cerebral cortex, showing homogeneous staining. Therefore, we did not find areas with significantly different Klotho content. However, the meaning of this expression in sections other than the frontal cortex is beyond the scope of our work.
Is this a model of chronic kidney disease or a model of acute kidney disease? This question is relevant since adenine causes kidney damage in a temporal manner.
We regret whether this point has not sufficiently clarified in the text. We used an experimental model of chronic kidney disease, and for this purpose, the enriched-adenine diet was continuously given to the animals for six weeks, without removing it at any time. In fact, to ensure the appropriate development of chronic kidney disease, as a marker of renal function, plasma creatinine was monitored weekly, albeit these data were not included on the paper. The levels of creatinine had a trend to steadily increase throughout the study, as it is shown in the following table:
|
Plasma creatinine (mg/dl) |
Control group |
CKD group |
|
Week 0 (baseline) |
0.626 ± 0.015 |
0.635 ± 0.012 |
|
Week 1 |
0.549 ± 0.073 |
0.820 ± 0.054* |
|
Week 2 |
0.563 ± 0.014 |
0.689 ± 0.050* |
|
Week 3 |
0.533 ± 0.032 |
0.724 ± 0.035* |
|
Week 4 |
0.481 ± 0.013 |
0.936 ± 0.060* |
|
Week 5 |
0.467 ± 0.067 |
1.092 ± 0.077** |
|
Week 6 |
0.768 ± 0.053 |
2.083 ± 0.277* |
Data expressed as mean ± SEM. *P<0.05 vs. Control; **P<0.01 vs. Control.
In addition, at week 6, the CKD group had significantly higher circulating levels of PTH in comparison with the group of rats with normal renal function (229 ± 17 vs. 984 ± 348 pg/ml, figure 1E). This increase in PTH is indicative of the onset of secondary hyperparathyroidism, which develops as a consequence of the progression of chronic kidney disease.
Altogether, because of the age of the rats, the six-week study period, the data from the table reflecting weekly changes in creatinine levels, and the evidence of secondary hyperparathyroidism, we consider the established renal damage to be chronic rather than acute. However, from the perspective of cognitive impairment this could be different. Although the progression of cognitive decline in the adenine-fed rat model has not been studied in depth, it could be considered that these cerebral lesions are initial or they are in early stage. It should be noted that for this cognitive impairment to occur, renal damage must first be established, so we presume that the changes in cognitive function occur after the onset of chronic in renal damage.
Hence, the following text in the discussion has been modified accordingly (p. 6-7): “Chronic kidney disease was induced by the administration of a diet with high content of adenine. […]This diet has proven to induce structural and functional changes, which mimics many of the features observed in human renal disease, including those related to the disturbances in mineral homeostasis and the development of secondary hyperparathyroidism [14]”.
Minor: Please improve image quality.
We appreciate the feedback, which will help improve the visualization of our results. We have noticed that after copy/paste the image from PowerPoint into the Word document, we lost the original brightness and sharpness of the image. After converting the image to jpg format using the GIMP-2.10 program, we achieved to maintain the original image quality. We hope this reedited version has improved and meets the standards for publication.

Round 2
Reviewer 1 Report
Comments and Suggestions for Authors
The author thoroughly answered all my queries, and their paper made some upgrades on the known knowledge base. They obtained further data on the timeframe and mechanisms. Also, I appreciate their revisions to the paper, which made these aspects more prominent. Therefore, I agree to its publication.
Reviewer 2 Report
Comments and Suggestions for Authors
Dear authors congratulations for this improved version of your work. I have no further comments.